# Asymmetric response of Northern Hemisphere near-surface wind speed to

| 2  | CO <sub>2</sub> removal                                                                                                                                                 |
|----|-------------------------------------------------------------------------------------------------------------------------------------------------------------------------|
| 3  |                                                                                                                                                                         |
| 4  | Zhi-Bo Li <sup>1</sup> , Chao Liu <sup>2</sup> , Cesar Azorin-Molina <sup>3</sup> , Soon-Il An <sup>4</sup> , Yang Zhao <sup>5,6</sup> , Yang Xu <sup>7</sup> , Jongsoo |
| 5  | Shin <sup>8</sup> , Deliang Chen <sup>9</sup> , Cheng Shen <sup>1,*</sup>                                                                                               |
| 6  |                                                                                                                                                                         |
| 7  | <sup>1</sup> Regional Climate Group, Department of Earth Sciences, University of Gothenburg,                                                                            |
| 8  | Gothenburg, Sweden                                                                                                                                                      |
| 9  | <sup>2</sup> School of Earth and Environmental Sciences, Seoul National University, Seoul, South                                                                        |
| 10 | Korea                                                                                                                                                                   |
| 11 | <sup>3</sup> Centro de Investigaciones sobre Desertificación, Consejo Superior de Investigaciones                                                                       |
| 12 | Científicas (CIDE, CSIC-UV-Generalitat Valenciana), Climate, Atmosphere and Ocean                                                                                       |
| 13 | Laboratory (Climatoc-Lab), Moncada, Valencia, Spain                                                                                                                     |
| 14 | <sup>4</sup> Department of Atmospheric Sciences, Yonsei University, Seoul, South Korea                                                                                  |
| 15 | <sup>5</sup> Frontiers Science Center for Deep Ocean Multispheres and Earth System-                                                                                     |
| 16 | Key Laboratory of Physical Oceanography-Institute for Advanced Ocean Studies-                                                                                           |
| 17 | Academy of the Future Ocean, Ocean University of China, Qingdao, China                                                                                                  |
| 18 | <sup>6</sup> College of Oceanic and Atmospheric Sciences, Ocean University of China, Qingdao, China                                                                     |
| 19 | <sup>7</sup> Department of Atmospheric Science, Yunnan University, Kunming, China                                                                                       |
| 20 | <sup>8</sup> Woods Hole Oceanographic Institution, Woods Hole, MA, USA                                                                                                  |
| 21 | <sup>9</sup> Department of Earth System Sciences, Tsinghua University, Beijing, China                                                                                   |
| 22 |                                                                                                                                                                         |
| 23 |                                                                                                                                                                         |
| 24 | *Corresponding author: Cheng Shen ( <a href="mailto:cheng.shen@gu.se">cheng.shen@gu.se</a> )                                                                            |
| 25 |                                                                                                                                                                         |
| 26 | Key words:                                                                                                                                                              |
| 27 | CO2 removal; AMOC; near-surface wind; European wind; Asymmetric response; Northern                                                                                      |
| 28 | hemisphere                                                                                                                                                              |
| 29 |                                                                                                                                                                         |

#### Abstract

Understanding changes in near-surface wind speed (NSWS) is crucial for weather extremes prediction and wind energy management. This study examines the response of NSWS to atmospheric carbon dioxide (CO<sub>2</sub>) removal using large ensemble simulations of the Community Earth System Model version 1.2 (CESM1.2) and the models participating in the Carbon Dioxide Removal Model Intercomparison Project. Our results reveal that increasing CO<sub>2</sub> concentrations lead to an overall weakening in the Northern Hemisphere (NH) extratropical NSWS over land. During the initial stage of CO2 removal (early ramp-down period), NH NSWS rapidly recovers. However, this recovery stalls and transitions into a declining trend during the late ramp-down period, mainly driven by pronounced negative NSWS trends in Europe. We find that a concurrent rapid recovery of the Atlantic Meridional Overturning Circulation (AMOC) counteracts the global cooling-induced recovery of the North Atlantic meridional air temperature gradient and associated westerly jet, thus prolonging NSWS weakening in NH mid-latitudes. Our findings underscore the pivotal and phasedependent role of AMOC in regulating NH extratropical NSWS variability under varying CO<sub>2</sub> concentrations, offering valuable insights for future climate adaptation strategies.

#### 1 Introduction

The phenomena of terrestrial near-surface wind speed (~10 m above the ground; NSWS) "stilling" and subsequent "reversal" have been recognized for over a decade, yet significant gaps remain in understanding their underlying mechanisms (Wu et al., 2018; Zeng et al., 2019). Accurate future projections of NSWS have garnered significant attention primarily due to their

and Li, 2020; Pryor et al., 2021; Shen et al., 2024). Several studies based on climate model simulations have investigated future spatiotemporal variations of NSWS in response to increasing atmospheric carbon dioxide (CO<sub>2</sub>), revealing complex global and regional responses influenced by polar amplification and altered land-sea thermal gradients (Bichet et al., 2012; Karnauskas et al., 2018; Shen et al., 2021; Zha et al., 2021; Deng et al., 2022). Future projections using these models, mostly from the Coupled Model Intercomparison Project Phases 5 and 6 (CMIP5 and CMIP6) (Taylor et al., 2012; O'Neill et al., 2016), consistently project robust reductions in NSWS over mid-latitude land areas of the Northern Hemisphere (NH) and increases in parts of the Southern Hemisphere by the end of the 21st century (Karnauskas et al., 2018; Zha et al., 2021; Deng et al., 2022; Shen et al., 2022). However, these idealized CO<sub>2</sub> experiments typically focus on scenarios involving continuously rising CO<sub>2</sub> concentrations through the 21st century, while the potential impacts of subsequent CO<sub>2</sub> removal on NSWS have not been examined. Clarifying this response is crucial, given global decarbonization objectives and the anticipated widespread deployment of wind energy resources (Lei et al., 2023). The irreversibility and asymmetry of various climate phenomena have been investigated through CO<sub>2</sub> ramp-up and ramp-down experiments using the global climate models. Many

implications for wind energy development (Karnauskas et al., 2018; Zeng et al., 2019; Zhang

through CO<sub>2</sub> ramp-up and ramp-down experiments using the global climate models. Many studies have employed the standard CMIP "1pctCO<sub>2</sub>" experiment as the ramp-up experiment, in which CO<sub>2</sub> concentration gradually increases at a rate of 1% per year for 140 years until it quadruples relative to pre-industrial levels (Eyring et al., 2016). On this basis, different ramp-down scenarios have been performed to assess the reversibility of CO<sub>2</sub>-induced climate change

(Wu et al., 2010; Cao et al., 2011; Boucher et al., 2012; MacDougall, 2013; Wu et al., 2014; 74 Ma et al., 2016; Field and Mach, 2017; Ehlert and Zickfeld, 2018). To systematically explore 75 such scenarios, the Carbon Dioxide Removal Model Intercomparison Project (CDRMIP) in 76 CMIP6 has been launched to provide 1pctCO<sub>2</sub>-carbon dioxide removal experiment as the 77 ramp-down period, in which the climate initiates from the end of the 1pctCO<sub>2</sub> experiment and 78 the evolution of CO<sub>2</sub> concentration mirrors that in the 1pctCO<sub>2</sub> experiment (Keller et al., 2018). 79 These experiments facilitate evaluation of climate responses and the associated uncertainties 80 stemming from model differences (Zhang et al., 2023; Jin et al., 2024; Su et al., 2024). 81 Following the CDRMIP protocol (Keller et al., 2018), An et al. (2021) conducted large-82 ensemble simulations of CO<sub>2</sub> ramp-up and ramp-down simulations using the Community 83 Earth System Model version 1.2 (CESM1.2). It is found that major ocean circulation systems, 84 particularly the Atlantic Meridional Overturning Circulation (AMOC), shows a unique 85 delayed recovery that significantly shapes the responses of other climatic factors (An et al., 86 2021; Oh et al., 2022). Subsequent studies expanded on these results to examine implications 87 for the hydrological cycle (Yeh et al., 2021; Kim et al., 2022; Kug et al., 2022; Im et al., 2024), 88 El Niño-Southern Oscillation (Liu et al., 2023a, b; Pathirana et al., 2023), Hadley cell (Kim 89 et al., 2023), gross primary productivity (Yang et al., 2024) and mid-latitude storm tracks 90 (Hwang et al., 2024), all demonstrating varying degrees of irreversibility. Regarding wind 91 speed, Hwang et al. (2024) reported enhanced cyclone-related surface wind extremes in 92 93 southern North America and Europe during the late CO2 removal period. However, the responses of mean NSWS under these CO<sub>2</sub> scenarios have received little attention. 94

(Shen et al., 2022) and considering the importance of wind energy resources in the NH midlatitudes are expected to be significantly developed (Pryor et al., 2020). We are motivated to investigate how NSWS in the NH mid-latitudes would respond to potential future CO<sub>2</sub> removal. To ensure robustness, we analyze results from CESM1.2 alongside those from three available CMIP6 models participating in the CDRMIP.

# 2 Data and Method

#### 2.1 CESM1.2 Simulations

CESM1.2 (Hurrell et al., 2013) includes the atmosphere (Community Atmospheric Model version 5), ocean (Parallel Ocean Program version 2), sea ice (Community Ice Code version 4), and land models (Community Land Model version 4). The atmospheric component features a horizontal resolution of approximately 1° and 30 vertical levels (Neale et al., 2012). The ocean component includes 60 vertical levels, with a longitudinal resolution of about 1° and a latitudinal resolution of about 0.3° near the equator, increasing gradually to about 0.5° near the poles (Smith et al., 2010). The land component includes the carbon-nitrogen cycle (Lawrence et al., 2011).

The experiment followed idealized CO<sub>2</sub> scenarios in two phases (An et al., 2021). In the first phase, the CO<sub>2</sub> concentration was held constant at 367 ppm, representing present-day levels, and the model is integrated for 900 years. In the second phase, the CO<sub>2</sub> concentration increased from 367 ppm to 1,478 ppm at a rate of 1% per year over 140 years (2001–2140: ramp-up period), then decreased back to 367 ppm at the same rate over the next 140 years

(2141–2280: ramp-down period). After the ramp-down period, CO<sub>2</sub> levels were stabilized at 367 ppm for 220 years (2281–2500: stabilization period). This second phase was run with 28 ensemble members, each starting from different initial conditions taken from the present-day period, representing various phases of multi-decadal climate oscillations such as the Atlantic Multidecadal Oscillation and Pacific Decadal Oscillation. Such large ensemble simulations provide a sufficient tool to separate the forced responses from internal variability, making it effective in assessing forced changes in regional NSWS (Li et al., 2019; Deser et al., 2020; Zha et al., 2021).

#### 2.2 CDRMIP Simulations

We further utilized CMIP6 models in the CDRMIP to verify the results of CESM1.2. Notably, CanESM5, MIROC-ES2L, and NorESM2-LM models are the only three available models with variables of "sfcWind" (near-surface wind speed) and "msftmz" (stream function, for calculating AMOC), and each model contains one realization. We compared terrestrial NSWS (60°S–70°N) climatology in their present-day run (CESM1.2) and piControl experiments (CanESM5, MIROC-ES2L, and NorESM2-LM) with the ERA5 (Hersbach et al., 2020), and found that the pattern correlation coefficients are 0.85, 0.81, 0.72, and 0.89 for CESM1.2, CanESM5, MIROC-ES2L, and NorESM2-LM, respectively. For the 20°N–70°N terrestrial NSWS, the pattern correlation coefficients are 0.71, 0.59, 0.52, and 0.83, respectively (Figure S1). The magnitudes between four models and the ERA5 are comparable, and the area weighted root mean squared difference between models and ERA5 are 1.03, 1.07, 1.37, and 0.71 m s<sup>-1</sup>, respectively. These indicate an overall reasonable ability of models in

reproducing the large-scale characteristics of terrestrial NSWS.

#### 2.3 AMOC Index

The AMOC index was calculated as the maximum stream function value at 26.7°N in the North Atlantic (0°N–70°N, 60°W–30°E), providing a robust measure of AMOC strength and variability across different simulation phases (An et al., 2021).

#### 2.4 Statistical Methods

To facilitate consistent analysis, all data were bi-linearly interpolated to a uniform 1.5° × 1.5° grid. All calculations related to NSWS are focused on the land regions. An 11-year running mean was used for all physical variables, unless otherwise stated. To clarify and quantify the contributions of global-mean surface air temperature and AMOC to NSWS changes, we performed a bi-regression analysis. This analysis allowed us to assess the proportion of NSWS variance explained by each factor during different periods.

#### 3 NSWS Responses to CO<sub>2</sub> Ramp-up and Ramp-down

In the CESM1.2 simulations, the annual global-mean surface air temperature (GMST) increases by about 5°C from 2000 to 2140 (Figure 1a). This warming trend reverses during the ramp-down period (2141–2280) as CO<sub>2</sub> concentrations decrease, although the cooling trend is less pronounced than the preceding warming. During the stabilization period (2281–2500), GMST remains approximately 1°C above the year 2000 levels for roughly 40 years before gradually declining toward 2500. Spatially, the ramp-up period shows pronounced warming over high-latitude land areas, whereas oceanic warming is moderate (Figure S2a). The

Subpolar North Atlantic (SNA) exhibits minimal warming, known as "warming hole" or "cold blob" (Chemke et al., 2020; Keil et al., 2020; Rahmstorf, 2024), and is likely due to reduced heat transport associated with a weakening AMOC under global warming (Zhang et al., 2019). Conversely, during the ramp-down period (2141–2280), most areas cool, including the SNA (Figure S2b). The asymmetric surface air temperature (SAT) trends over the SNA are largely driven by the delayed recovery of AMOC during the late ramp-down period (2221–2280), which is influenced by increased salt advection feedback due to changes in the subtropic-to-subpolar salinity gradient and ocean stratification (An et al., 2021; Oh et al., 2022).

Throughout the CO2 ramp-up period (2001–2140), the NH-averaged (20°N–70°N)

annual-mean NSWS decreased, consistent with projections by CMIP6 models (Zha et al., 2021; Shen et al., 2022). The ramp-down period is divided into early (2141–2220) and late (2221–2280) ramp-down periods (An et al., 2021). During the early ramp-down period (2141–2220), NSWS in the NH extratropics rebounds rapidly to year-2000 levels at about double the rate of the ramp-up period. Correspondingly, SAT pattern features stronger cooling over the SNA (Figure S2c), similar to the ramp-up period, indicating a strengthened meridional SAT gradient. In contrast, during the late ramp-down period (2221–2280), NH extratropical NSWS changes moderately (Figure 1b), accompanied by a notable warming trend over the SNA and weaker cooling trends at high NH latitudes compared to the early ramp-down period (2141–2220) (Figures S2c–d). Throughout the stabilization period (2281–2500), NH NSWS initially continues to decline slowly, then gradually increases towards the end of the simulation.

To further examine the regional reversibility of NSWS under varying CO<sub>2</sub> conditions, Figures 2a–d show the spatial patterns of NH NSWS trends during the ramp-up, ramp-down,

early ramp-down, and late ramp-down periods, respectively. During the ramp-up period (2001–2140) (Figure 2a), significant negative NSWS trends prevail across mid-latitudes in the NH. During the ramp-down period (2141–2280), this spatial pattern reverses (Figure 2b), with a pattern correlation coefficient of -0.9 (P<0.01) between the ramp-up and ramp-down periods, highlighting the pronounced effect of CO<sub>2</sub> forcing on NSWS distributions. Spatial discrepancies between two ramp-down periods are primarily observed over the Eurasia continent: during the early ramp-down period (2141–2220), NSWS shows marked positive trends (Figure 2c); however, these positive trends turn negative in Europe and diminish in North America and Central Asia in the late ramp-down period (2221–2280) (Figure 2d).

Additionally, we validated NSWS responses using three CMIP6 models from CDRMIP (Figures S3 and S4). It is found that the time series of NSWS over the NH extratropics are generally similar across the three models (Figure S3), as are the spatial patterns of NSWS trends (Figure S4). Nevertheless, noticeable inter-model discrepancies exist between the CDRMIP models and the CESM1.2. A fast recovery of NSWS is observed in the NH extratropics in the CESM1.2 during the early ramp-down period (Figure 1b), while it shows overall symmetric changes in three CDRMIP models throughout ramp-up and ramp-down periods (Figure S3). These differences likely stem from differing AMOC evolutions among the models, a point we elaborate further in the following section.

### 4 Effect of AMOC on Modulating the NH Extratropical NSWS

Previous studies indicate that hemispheric-scale NSWS changes are strongly influenced by large-scale meridional SAT gradient (Zha et al., 2021; Deng et al., 2022; Shen et al., 2022;

Li et al., 2024), with NH mid-latitude NSWS closely linked to variations in the westerly jet due to vertical downward momentum transport from upper tropospheric levels (Shen et al., 2023; Shen et al., 2025). Moreover, the strength of the AMOC critically affects these regional patterns by modulating the meridional SAT gradient and the westerly jet through its control of the SNA temperature (Zhang et al., 2019; An et al., 2021; Hwang et al., 2024). The CESM1.2 simulations reveal a clear weakening trend in AMOC (~0.5 Sv decade<sup>-1</sup>) during the ramp-up period extending until about year 2200, followed by a rapid recovery (~1.2 Sv decade<sup>-1</sup>) persisting until around year 2300 (Figure 3a). By comparison, the three CDRMIP models exhibit symmetric declining and recovering trends in AMOC strength during ramp-up and ramp-down periods, respectively (Figure S5). The different responses of AMOC across models significantly affect regional SAT gradient, thereby altering regional atmospheric circulation (Hwang et al., 2024).

To elucidate the combined effects of CO<sub>2</sub> levels and AMOC variability on NSWS, we analyzed temporal evolutions of meridional SAT gradients and westerly jets for both NH and North Atlantic (NA) (Figures 3b–e). The NH meridional SAT gradient was defined as the SAT difference between tropical ( $0^{\circ}N-30^{\circ}N$ ) and high-latitude ( $60^{\circ}N-90^{\circ}N$ ) bands. Mid-latitude westerly jets were defined as the average 500 hPa zonal winds for  $30^{\circ}N-60^{\circ}N$ . A significant negative correlation (-0.78, P<0.01) exists between the NH extratropical NSWS (Figure 1b) and the NH SAT gradient (Figure 3b), whereas a strong positive correlation (0.91, P<0.01) is detected with the NH westerly jet (Figure 3c). During the ramp-up period (2001-2140), reductions in the NH SAT gradient contribute to a weaker westerly jet and reduced NSWS across extratropics. Conversely, in the early ramp-down period (2141-2220), NSWS quickly

rebounds alongside recovery of the SAT gradient and westerly jet. However, during the late ramp-down period (2221–2280), the NH SAT gradient stabilizes (Figure 3b), resulting in slow changes in both the NH westerly jet (Figure 3c) and extratropical NSWS (Figure 1b), despite the CO<sub>2</sub>-removal-induced global cooling. These dynamics underscore an effect from the AMOC, which strongly modulates the SAT gradient in the NA, and thus local NSWS trends. During the ramp-up period (2001–2140), the NA SAT gradient (60°W–30°E, 0°N–30°N minus 60°W–30°E, 60°N–90°N) weaken by approximately 1°C (Figure 3d), a relatively small magnitude compared to the 6.5°C reduction in the NH SAT gradient (Figure 3b). The NA SAT gradient changes are modulated by two combined effects: CO2-induced global warming/cooling and the AMOC strength (Zhang et al., 2019). This milder NA SAT gradient change is related to a weakened AMOC's role in transporting warm water to the SNA, which partially offsets the hemisphere-scale SAT gradient weakening induced by global warming (Figure 3d). And the weakened NA SAT gradient favors a weakened NA NSWS (Fig. 1b). In the early ramp-down period (2141-2220), the synergistic effects of global cooling and a further weakened AMOC promote a stronger enhancement of the NA SAT gradient (Figure 3d) and westerly jet (Figure 3e), which induce a rapidly enhancement of NA NSWS (Fig. 1b). Conversely, during the late ramp-down period (2221–2280), the rapid recovery of the AMOC dramatically increases warm water flow to the SNA (Figure S1d), diminishes the NA SAT gradient (Figure 3d) despite the prevailing global cooling effect from CO<sub>2</sub>-removal, and leads to a weakened NA westerly jet (Figure 3e) and NSWS (Fig. 1b).

Spatial patterns of 500 hPa zonal winds (Figure S6) reflect similar tendencies to those of NSWS in the NH, particularly across mid-latitudes where enhanced zonal winds correlate with

increased NSWS (Figure 2), and vice versa. Throughout the ramp-up period (2001–2140), there is a general weakening trend of westerly jets over the Asian and North American continents, while regional westerlies intensify over NA and Europe (Figure S6a). The early ramp-down period (2141–2220) witnesses significant strengthening trend of NH westerly jets (Figure S6c), propelled by global cooling and a weakened AMOC (Figure S2c). During the late ramp-down period (2221–2280), recovery of the AMOC causes significant weakening of westerly jets over NA and Europe by reducing the NA SAT gradient (Zhang et al., 2019; An et al., 2021; Hwang et al., 2024) (Figures 3d–e), which correspondingly weakens NSWS over Europe (Figure 2d).

Internally generated AMOC changes can also support our argument about the potential roles played by CO<sub>2</sub>-forced AMOC changes. To quantify how internal AMOC variability contributes to the cross-member spread of key NA climate variables, Figure 4 shows the yearly Pearson correlation coefficients among the 28 CESM ensemble members between AMOC strength and (i) the NA meridional SAT gradient, (ii) the NA westerly-jet intensity, and (iii) the European (30°–60°N, 5°W–60°E) NSWS. Because all members share identical external forcing, these inter-ensemble correlations isolate internal variability. Significant correlations thus indicate that internal AMOC fluctuations are major drivers of atmospheric variability among ensemble members. During the ramp-up period (2001–2140), the ensemble spread of AMOC remains small, and correlations with the three metrics are weak and statistically insignificant. However, during the late ramp-down period (2221–2280), the spread of AMOC substantially increases, and the correlations become statistically significant. These year-by-year correlations clearly demonstrate that larger (smaller) AMOC anomalies are associated

with a weaker (stronger) NA SAT gradient, a reduced (enhanced) westerly jet, and lower (higher) European NSWS.

Moreover, we conducted a bi-regression analysis using GMST and AMOC as explanatory variables for NH extratropical NSWS variability across three distinct periods. During the ramp-up period (2001–2140), GMST and AMOC explain the variance of NH extratropical NSWS for 98% and 1.2%, respectively. During the early ramp-down period (2141–2220), GMST and AMOC explain 95.3% and 4.2% of the variance, respectively. However, during the late ramp-down period (2221–2280), the relative influence of AMOC sharply increases, accounting for 73.2% of variance, surpassing GMST (24.5%). Thus, while GMST predominantly governs NH extratropical NSWS changes during periods of weaker AMOC variability, AMOC emerges as the dominant factor only when its anomalies become significantly large, such as during its rapid recovery in the late ramp-down phase. Besides, we also performed similar analysis for the three CDRMIP models. Among three models and two periods, the contributions of GMST always dominate (contributions >90%) the NH extratropical NSWS changes, further supporting that a substantial AMOC anomaly is required for AMOC to notably influence NSWS changes, as observed in CESM.

# **5 Summary and Outlook**

In this study, we utilized a CESM1.2 CO<sub>2</sub> removal experiment to evaluate the response of NH extratropical NSWS to anthropogenic CO<sub>2</sub> emission and subsequent removal. Our analyses reveal an asymmetric response of NSWS during the ramp-up (2001–2140) and ramp-down (2141–2280) periods, driven by phase-dependent interactions between GMST changes

and AMOC-related heat transport anomalies. Figure 5 summarizes the underlying physical mechanisms: During the ramp-up period (Figure 5a), the gradually weakening AMOC transports less warm water northward, thereby enhancing the meridional SAT gradient and thus strengthen NH and NA NSWS. However, the counteracting effect of CO<sub>2</sub>-driven global warming strongly reduces the NH meridional SAT gradient, ultimately dominating the response and resulting in decreased NSWS across NH extratropics. During the early ramp-down period (Figure 5b), CO<sub>2</sub> removal-induced global cooling combines with further AMOC weakening to substantially enhance the SAT gradient, intensifying the westerly jet and increasing NH NSWS, with GMST remaining the dominant driver due to its stronger anomalies. Conversely, during the late ramp-down period (Figure 5c), a rapid AMOC recovery substantially increases northward heat transport, significantly weakening the NA SAT gradient and subsequently diminishing the westerly jet and NSWS. This AMOC-driven weakening outweighs concurrent weaker GMST cooling-induced strengthening of the SAT gradient, inducing a weakening NH and NA NSWS.

The European NSWS exhibits distinct anticlockwise trajectories in relation to CO<sub>2</sub> concentrations (Figure 6a), signifying stronger NSWS during CO<sub>2</sub> removal than during the ramp-up period at identical CO<sub>2</sub> levels (Kim et al., 2022). Although there are also asymmetries of NSWS over Aisa and North America (Figures 6b–c), their magnitudes are substantially weaker compared to Europe. These phenomena imply the particular importance of AMOC variability for European NSWS. Furthermore, three CDRMIP models generally replicate NH extratropical NSWS trends but fail to capture the pronounced asymmetry found in CESM, likely due to their relatively symmetric AMOC evolutions, which yield almost linear NSWS

AMOC dynamics. We acknowledge several uncertainties inherent in our findings, including potential model-dependency arising from CESM's large ensemble experiment and limited realizations from the CDRMIP simulations. Future studies involving more extensive multimodel ensembles could further validate and enhance the robustness of our conclusions.

Before this study, the impact of the AMOC on terrestrial NSWS changes had been less recognized. Our findings advance the understanding within the wind research community by highlighting a crucial role for ocean circulation in modulating long-term NSWS trends. The enhanced NH NSWS during the early CO<sub>2</sub> removal period could significantly increase wind energy production (Pryor et al., 2020) but also heighten the risks of wind-related extremes (Hwang et al., 2024; Yu et al., 2024). Consequently, it is critical for monitoring efforts and climate policy formulations to consider these findings in their long-term climate strategies to optimize benefits and mitigate risks associated with wind energy and climate interventions.

## **Declaration of competing interests**

The authors declare no conflicts of interest.

#### Acknowledgments

This work is supported by the Swedish Formas (2023-01648, 2020-00982) and the Swedish Research Council (VR: 2021-02163). C.A-M. is also supported by the CSIC Interdisciplinary Thematic Platform (PTI) "PTI+ Clima y Servicios Climáticos" and the RED-CLIMA 2 (ref. LINCG24042). S.-I.A. is also supported by National Research Foundation of Korea grants funded by the Korean government (NRF-2018R1A5A1024958). C.S. is also supported by the

| 335                      | Sven Lindqvists Forskningsstiftelse, Stiftelsen Längmanska kulturfonden (BA24-0484) and                                                                                                                                                                                                                                                                                                                |
|--------------------------|--------------------------------------------------------------------------------------------------------------------------------------------------------------------------------------------------------------------------------------------------------------------------------------------------------------------------------------------------------------------------------------------------------|
| 336                      | Adlerbertska Forskningsstiftelsen (AF2024-0069). The CESM simulation was conducted on                                                                                                                                                                                                                                                                                                                  |
| 337                      | the supercomputer supported by the National Center for Meteorological Supercomputer of                                                                                                                                                                                                                                                                                                                 |
| 338                      | Korea Meteorological Administration (KMA), the National Supercomputing Center with                                                                                                                                                                                                                                                                                                                     |
| 339                      | supercomputing resources, associated technical support (KSC-2021-CHA-0030), and the                                                                                                                                                                                                                                                                                                                    |
| 340                      | Korea Research Environment Open NETwork (KREONET).                                                                                                                                                                                                                                                                                                                                                     |
| 341                      | Data Availability                                                                                                                                                                                                                                                                                                                                                                                      |
| 342                      | All data used in this study are publicly available. The CESM1.2 outputs are available from                                                                                                                                                                                                                                                                                                             |
| 343                      | https://data.mendeley.com/datasets/f5ry6hgxkw/2. CMIP6 outputs are available from the                                                                                                                                                                                                                                                                                                                  |
| 344                      | Earth System Grid Federation ( <a href="https://esgf-data.dkrz.de/projects/esgf-dkrz/">https://esgf-data.dkrz.de/projects/esgf-dkrz/</a> ). And ERA5                                                                                                                                                                                                                                                   |
| 345                      | reanalysis data are available from the European Centre for Medium-Range Weather Forecasts                                                                                                                                                                                                                                                                                                              |
| 346                      | (https://cds.climate.copernicus.eu/cdsapp#!/dataset/reanalysis-era5-single-                                                                                                                                                                                                                                                                                                                            |
| 347                      | levels?tab=overview).                                                                                                                                                                                                                                                                                                                                                                                  |
| 348                      | Supplementary Information                                                                                                                                                                                                                                                                                                                                                                              |
| 349                      | The manuscript contains supplementary materials.                                                                                                                                                                                                                                                                                                                                                       |
| 350                      |                                                                                                                                                                                                                                                                                                                                                                                                        |
| 351                      | References                                                                                                                                                                                                                                                                                                                                                                                             |
| 352<br>353<br>354<br>355 | An, SI., J. Shin, SW. Yeh, SW. Son, JS. Kug, SK. Min, and HJ. Kim (2021). Global cooling hiatus driven by an AMOC overshoot in a carbon dioxide removal scenario, <i>Earth's Future</i> , <i>9</i> , e2021EF002165. <a href="https://doi.org/10.1029/2021EF002165">https://doi.org/10.1029/2021EF002165</a> Bichet, A., M. Wild, D. Folini, and C. Schar (2012). Causes for decadal variations of wind |
| 356<br>357               | speed over land: Sensitivity studies with a global climate model, <i>Geophysical Research Letters</i> , 39(11). <a href="https://doi.org/10.1029/2012GL051685">https://doi.org/10.1029/2012GL051685</a>                                                                                                                                                                                                |
| 358                      | Boucher, O., P. R. Halloran, E. J. Burke, M. Doutriaux-Boucher, C. D. Jones, J. Lowe, M. A.                                                                                                                                                                                                                                                                                                            |
| 359                      | Ringer, E. Robertson, and P. Wu (2012). Reversibility in an Earth System model in                                                                                                                                                                                                                                                                                                                      |
| 360                      | response to CO2 concentration changes, Environmental Research Letters, 7, 024013.                                                                                                                                                                                                                                                                                                                      |

https://doi.org/10.1088/1748-9326/7/2/024013

- Cao, L., G. Bala, and K. Caldeira (2011). Why is there is a short-term increase in global precipitation in response to diminished CO<sub>2</sub> forcing? *Geophysical Research Letters*, 38, L06703. https://doi.org/10.1029/2011GL046713
- Chemke, R., L. Zanna, and L. M. Polvani (2020). Identifying a human signal in the North Atlantic warming hole. *Nature Communications*, 11(1), 1–7. https://doi.org/10.1038/s41467-020-15285-x
- Deng, K., W. Liu, C. Azorin-Molina, S. Yang, H. Li, G. Zhang, L. Minola, and D. Chen (2022).

  Terrestrial stilling projected to continue in the Northern Hemisphere mid-latitudes,

  Earth's Future, 10(7), e2021EF002448. https://doi.org/10.1029/2021EF002448
- Deser, C., F. Lehner, K. B. Rodgers, T. Ault, T. L. Delworth, P. N. DiNezio, A. Fiore, C. Frankignoul, J. C. Fyfe, and D. E. Horton (2020). Insights from Earth system model initial-condition large ensembles and future prospects, *Nature Climate Change*, *10*(4), 277–286. <a href="https://doi.org/10.1038/s41558-020-0731-2">https://doi.org/10.1038/s41558-020-0731-2</a>
- Ehlert, D, and K. Zickfeld (2018). Irreversible ocean thermal expansion under carbon dioxide removal, *Earth System Dynamics*, *9*(1), 197–210. <a href="https://doi.org/10.5194/esd-9-197-2018">https://doi.org/10.5194/esd-9-197-2018</a>
- Eyring, V., S. Bony, G. A. Meehl, C. A. Senior, B. Stevens, R. J. Stouffer, and K. E. Taylor (2016). Overview of the Coupled Model Intercomparison Project Phase 6 (CMIP6) experiment design and organization, *Geoscientific Model Development*, 9, 1937–1958. <a href="https://doi.org/10.5194/gmd-9-1937-2016">https://doi.org/10.5194/gmd-9-1937-2016</a>
- Field, C. B., and K. J. Mach (2017). Rightsizing carbon dioxide removal, *Science*, *356*(6339), 706–707. <a href="https://doi.org/10.1126/science.aam9726">https://doi.org/10.1126/science.aam9726</a>
- Hersbach, H., B. Bell, P. Berrisford, S. Hirahara, A. Horányi, J. Muñoz-Sabater, J. Nicolas, C. Peubey, et al. (2020). The ERA5 global reanalysis, *Quarterly Journal of the Royal Meteorological Society*, 146(730), 1999-2049. https://doi.org/10.1002/qj.3803
- Hurrell, J. W., M. M. Holland, P. R. Gent, S. Ghan, J. E. Kay, and P. J. Kushner (2013). The community earth system model: A framework for collaborative research. *Bulletin of the American Meteorological Society*, *94*(9), 1339–1360. <a href="https://doi.org/10.1175/BAMS-D-12-00121.1">https://doi.org/10.1175/BAMS-D-12-00121.1</a>
- Hwang, J., S.-W. Son, C. I. Garfinkel, T. Woollings, H. Yoon, S.-I. An, S.-W. Yeh, S.-K., Min,
   J.-S. Kug, and J. Shin (2024). Asymmetric hysteresis response of mid-latitude storm
   tracks to CO<sub>2</sub> removal, *Nature Climate Change*, 14, 496–503.
   <a href="https://doi.org/10.1038/s41558-024-01971-x">https://doi.org/10.1038/s41558-024-01971-x</a>
- Im, N., D., Kim, S.-I., An, S., Paik, S.-K., Kim, J., Shin, S.-K., Min, et al. (2024). Hysteresis of European summer precipitation under a symmetric CO<sub>2</sub> ramp-up and ramp-down pathway, *Environmental Research Letters*, 19, 074030. <a href="https://doi.org/10.1088/1748-9326/ad52ad">https://doi.org/10.1088/1748-9326/ad52ad</a>
- Jin, J., D. Ji, X. Dong, K. Fei, R. Guo, J. He, Y. Yu, et al. (2024). CAS-ESM2.0 dataset for the Carbon Dioxide Removal Model Intercomparison Project (CDRMIP), *Advances in Atmospheric Sciences*, 41, 989–1000. <a href="https://doi.org/10.1007/s00376-023-3089-3">https://doi.org/10.1007/s00376-023-3089-3</a>
- Karnauskas, K. B., J. K. Lundquist, and L. Zhang (2018). Southward shift of the global wind energy resource under high carbon dioxide emissions, *Nature Geoscience*, *11*(1), 38–43. https://doi.org/10.1038/s41561-017-0029-9

- Keil, P., T. Mauritsen, J. Jungclaus, C. Hedemann, D. Olonscheck, and R. Ghosh (2020).

  Multiple drivers of the North Atlantic warming hole. *Nature Climate Change*, 10(7), 667–671. https://doi.org/10.1038/s41558-020-0819-8
- Keller, D. P., A. Lenton, V. Scott, N. E. Vaughan, N. Bauer, D. Ji, C. D. Jones, et al. (2018).
   The Carbon Dioxide Removal Model Intercomparison Project (CDRMIP): rationale and
   experimental protocol for CMIP6, *Geoscientific Model Development*, 11(3), 1133–1160.
   https://doi.org/10.5194/gmd-11-1133-2018
- Kim, S.-K., J. Shin, S.-I. An, H.-J. Kim, N. Im, S. Xie, J.-S. Kug, and S.-W. Yeh (2022).
  Widespread irreversible changes in surface temperature and precipitation in response to
  CO<sub>2</sub> forcing, *Nature Climate Change*, 12, 834–840. <a href="https://doi.org/10.1038/s41558-022-01452-z">https://doi.org/10.1038/s41558-022-01452-z</a>
- Kim, S.-Y., Y.-J. Choi, S.-W. Son, K. M. Grise, P. W. Staten, S.-I. An, S.-W. Yeh, J.-S. Kug, S.-K. Min, and J. Shin (2023). Hemispherically asymmetric Hadley cell response to CO<sub>2</sub> removal, *Science Advances*, 9(30), eadg1801. <a href="https://doi.org/10.1126/sciadv.adg1801">https://doi.org/10.1126/sciadv.adg1801</a>
- Kug, J.-S., J.-H. Oh, S.-I. An, S.-W. Yeh, S.-K. Min, S.-W. Son, J. Kam, Y.-G. Ham, and J. Shin (2022). Hysteresis of the intertropical convergence zone to CO<sub>2</sub> forcing, *Nature Climate Change*, 12, 47–53. <a href="https://doi.org/10.1038/s41558-021-01211-6">https://doi.org/10.1038/s41558-021-01211-6</a>
- Lawrence, D. M., K. W. Oleson, M. G. Flanner, P. E. Thornton, S. C. Swenson, P. J. Lawrence, et al. (2011). Parameterization improvement sand functional and structural advances in Version 4 of the community land model. *Journal of Advances in Modeling Earth Systems*, 3(3), 1–27. <a href="https://doi.org/10.1029/2011ms000045">https://doi.org/10.1029/2011ms000045</a>
- Lei, Y., Z. Wang, D. Wang, X. Zhang, H. Che, X. Yue, C. Tian, J. Zhong, L. Guo, L. Li, H. Zhou, L. Liu, and Y. Xu (2023). Co-benefits of carbon neutrality in enhancing and stabilizing solar and wind energy, *Nature Climate Change*, 13, 693–700. <a href="https://doi.org/10.1038/s41558-023-01692-7">https://doi.org/10.1038/s41558-023-01692-7</a>
- Li, Z.-B., Y. Sun, T. Li, T. Hu, and Y. Ding (2019). Future changes in East Asian summer monsoon circulation and precipitation under 1.5 to 5 °C of warming, *Earth's Future*, 7(12), 1391–1406. https://doi.org/10.1029/2019EF001276
- Li, Z.-B., Y. Xu, H.-S. Yuan, Y. Chang, and C. Shen (2024). AMO footprint of the recent near-surface wind speed change over China, *Environmental Research Letters*, 19, 114031. <a href="https://doi.org/10.1088/1748-9326/ad7ee4">https://doi.org/10.1088/1748-9326/ad7ee4</a>
- Liu, C., S.-I. An, F. Jin, J. Shin, J.-S. Kug, W. Zhang, M. F. Stuecker, X. Yuan, A. Xue, X. Geng, and S.-K. Kim (2023a). Hysteresis of the El Niño–Southern Oscillation to CO<sub>2</sub> forcing, *Science Advances*, 9(31), eadh8442. <a href="https://doi.org/10.1126/sciadv.adh8442">https://doi.org/10.1126/sciadv.adh8442</a>
- Liu, C., S.-I. An, F. Jin, M. F. Stuecker, W. Zhang, J.-S. Kug, X. Yuan, J. Shin, A. Xue, X. Geng and S.-K. Kim (2023b). ENSO skewness hysteresis and associated changes in strong El Niño under a CO<sub>2</sub> removal scenario, *npj Climate and Atmospheric Science*, 6, 117. https://doi.org/10.1038/s41612-023-00448-6
- Ma, J., G. R. Foltz, B. J. Soden, G. Huang, J. He, and C. Dong (2016). Will surface winds
   weaken in response to global warming? *Environmental Research Letters*, 11, 124012.
   <a href="https://doi.org/10.1088/1748-9326/11/12/124012">https://doi.org/10.1088/1748-9326/11/12/124012</a>
- 447 MacDougall. A. H., (2013). Reversing climate warming by artificial atmospheric carbon-

- dioxide removal: Can a Holocene-like climate be restored? *Geophysical Research Letters*, 40(20), 5480–5485. https://doi.org/10.1002/2013GL057467
- Neale, R. B., C.-C. Chen, A. Gettelman, P. H. Lauritzen, S. Park, D. L. Williamson, et al. (2012). Description of the NCAR community atmosphere model (CAM 5.0). NCAR Tech. Note NCAR/TN-486+ STR (pp. 1–12). National Center for Atmospheric Research.
- Oh, J.-H., S.-I. An, J. Shin, and J.-S. Kug (2022). Centennial memory of the Arctic Ocean for future Arctic climate recovery in response to a carbon dioxide removal. *Earth's Future*, 10(8), e2022EF002804. <a href="https://doi.org/10.1029/2022EF002804">https://doi.org/10.1029/2022EF002804</a>
- O'Neill, B. C., C. Tebaldi, D. P. Van Vuuren, V. Eyring, P. Friedlingstein, G. Hurtt, R. Knutti, E. Kriegler, et al. (2016). The scenario model intercomparison project (ScenarioMIP) for CMIP6, Geoscientific Model Development, 9(9), 3461–3482. <a href="https://doi.org/10.5194/gmd-9-3461-2016">https://doi.org/10.5194/gmd-9-3461-2016</a>
- Pathirana, G., J.-H. Oh, W. Cai, S.-I. An, S.-K. Min, S.-Y. Jo, J. Shin, and J.-S. Kug (2023).

  Increase in convective extreme El Niño events in a CO<sub>2</sub> removal scenario, *Science Advances*, 9(25), eadh2412. <a href="https://doi.org/10.1126/sciadv.adh2412">https://doi.org/10.1126/sciadv.adh2412</a>
- Pryor, S., and R. J. Barthelmie (2021). A global assessment of extreme wind speeds for wind energy applications, *Nature Energy*, *6*(3), 268–276. <a href="https://doi.org/10.1038/s41560-020-00773-7">https://doi.org/10.1038/s41560-020-00773-7</a>
- Pryor, S., R. J. Barthelmie, M. S. Bukovsky, L. R. Leung, and K. Sakaguchi (2020). Climate change impacts on wind power generation, *Nature Reviews Earth & Environment*, *1*(12), 627–643. https://doi.org/10.1038/s43017-020-0101-7
- Rahmstorf, S (2024). Is the Atlantic overturing circulation approaching a tipping point?

  Oceanography, 37(3), 16–29. <a href="https://doi.org/10.5670/oceanog.2024.501">https://doi.org/10.5670/oceanog.2024.501</a>
- Shen, C., J. Zha, D. Zhao, J. Wu, W. Fan, M. Yang, and Z.-B. Li (2021). Estimating centennial-scale changes in global terrestrial near-surface wind speed based on CMIP6 GCMs, *Environmental Research Letters*, 16(8), 084039. <a href="https://doi.org/10.1088/1748-9326/ac1378">https://doi.org/10.1088/1748-9326/ac1378</a>
- Shen, C., J. Zha, Z.-B. Li, C. Azorin-Molina, K. Deng, L. Minola, and D. Chen (2022). Evaluation of global terrestrial near-surface wind speed simulated by CMIP6 models and their future projections, *Annals of the New York Academy of Sciences*, *1518*(1), 249–263. <a href="https://doi.org/10.1111/nyas.14910">https://doi.org/10.1111/nyas.14910</a>
- Shen, C., H. Yuan, Z.-B. Li, X. Yang, L. Minola, Y. Chang, and D. Chen (2023). March near-surface wind speed hiatus over China since 2011, *Geophysical Research Letters*, 50(15), e2023GL104230. https://doi.org/10.1029/2023GL104230
- Shen, C., Z.-B. Li, H. Yuan, Y. Yu, Y. Lei, and D. Chen (2024). Increases of offshore wind potential in a warming world, *Geophysical Research Letters*, 50(15), e2024GL109494. https://doi.org/10.1029/2024GL109494
- Shen, C., Z.-B. Li, F. Liu, H. W. Chen, and D. Chen (2025). A robust reduction in near-surface wind speed after volcanic eruptions: Implications for wind energy generation, *The Innovation*, 6(1), 100734. <a href="https://doi.org/10.1016/j.xinn.2024.100734">https://doi.org/10.1016/j.xinn.2024.100734</a>
- Smith, R., P. Jones, B. Briegleb, F. Bryan, G. Danabasoglu, J. Dennis, et al. (2010). The parallel ocean program (POP) reference manual ocean component of the community climate system model (CCSM) and community earth system model (CESM). LAUR-01853 (pp.

- 491 1–140).
- Su, X., G. Huang, L. Wang, and T. Wang (2024). Global drought changes and attribution under carbon neutrality scenario, *Climate Dynamics*, 62, 7851–7868. <a href="https://doi.org/10.1007/s00382-024-07310-2">https://doi.org/10.1007/s00382-024-07310-2</a>
- Taylor, K. E., R. J. Stouffer, and G. A. Meehl (2012). An overview of CMIP5 and the experiment design, *Bulletin of the American Meteorological Society*, *93*(4), 485–498. https://doi.org/10.1175/BAMS-D-11-00094.1
- Wu, J., J. Zha, D. Zhao, and Q. Yang (2018). Changes in terrestrial near-surface wind speed and their possible causes: an overview, *Climate Dynamics*, 51(5–6), 2039–2078. https://doi.org/10.1007/s00382-017-3997-y
- Wu, P., J. Ridley, A. Pardaens, R. Levine, and J. Lowe (2014). The reversibility of CO<sub>2</sub> induced climate change, *Climate Dynamics*, 45, 745–754. <a href="https://doi.org/10.1007/s00382-014-2302-6">https://doi.org/10.1007/s00382-014-2302-6</a>
- Wu, P., R. Wood, J. Ridley, and J. Lowe (2010). Temporary acceleration pf the hydrological cycle in response to a CO<sub>2</sub> rampdown, *Geophysical Research Letters*, *37*(12), L043730. https://doi.org/10.1029/2010GL043730
- Yang, Y.-M., J. Shin, S.-W. Park, J.-H. Park, S.-I. An, J.-S., Kug, S.-W. Yeh, et al. (2024). Fast reduction of Atlantic SST threatens Europe-wide gross primary productivity under positive and negative CO2 emissions. *npj Climate and Atmospheric Science*, 7(1), 117. https://doi.org/10.1038/s41612-024-00674-6
- Yeh, S.-W., S.-Y. Song, R. P. Allan, S.-I. An, and J. Shin (2021). Contrasting response of hydrological cycle over land and ocean to a changing CO<sub>2</sub> pathway, *npj Climate and Atmospheric Science*, 4, 53. https://doi.org/10.1038/s41612-021-00206-6
- Yu, Y., Z.-B. Li, Z. Yan, H. Yuan, and C. Shen (2024). Projected emergence seasons of year-maximum near-surface wind speed, *Geophysical Research Letters*, 51(2). https://doi.org/10.1029/2023GL107543
- Zeng, Z., A. D. Ziegler, T. Searchinger, L. Yang, A. Chen, K. Ju, S. Piao, L. Z. Li, P. Ciais, and D. Chen (2019). A reversal in global terrestrial stilling and its implications for wind energy production, *Nature Climate Change*, *9*(12), 979–985. <a href="https://doi.org/10.1038/s41558-019-0622-6">https://doi.org/10.1038/s41558-019-0622-6</a>
- Zha, J., C. Shen, Z.-B. Li, J. Wu, D. Zhao, W. Fan, M. Sun, C. Azorin-Molina, and K. Deng (2021). Projected changes in global terrestrial near-surface wind speed in 1.5° C–4.0° C global warming levels, *Environmental Research Letters*, 16(11), 114016. <a href="https://doi.org/10.1088/1748-9326/ac2fdd">https://doi.org/10.1088/1748-9326/ac2fdd</a>
- Zha, J., C. Shen, J. Wu, D. Zhao, W. Fan, H. Jiang, and T. Zhao (2023). Evaluation and projection of changes in daily maximum wind speed over China based on CMIP6.

  Journal of Climate, 36(5), 1503–1520. https://doi.org/10.1175/JCLI-D-22-0193.1
- Zhang, R., R. Sutton, G. Danabasoglu, Y. O. Kwon, R. Marsh, S. G. Yeager, D. E. Amrhein, and C. M. Little (2019). A review of the role of the Atlantic meridional overturning circulation in Atlantic multidecadal variability and associated climate impacts, *Reviews* of *Geophysics*, 57(2), 316–375. https://doi.org/10.1029/2019RG000644
- Zhang, S., and X. Li (2020). Future projections of offshore wind energy resources in China using CMIP6 simulations and a deep learning-based downscaling method, *Energy*, *217*,

| 534 | 119321. https://doi.org/10.1016/j.energy.2020.119321                                    |
|-----|-----------------------------------------------------------------------------------------|
| 535 | Zhang, S., X. Qu, G. Huang, and P. Hu (2023). Asymmetric response of South Asian summer |
| 536 | monsoon rainfall in a carbon dioxide removal scenario, npj Climate and Atmospheric      |
| 537 | Science, 6(1), 10. https://doi.org/10.1038/s41612-023-00338-x                           |
| 538 |                                                                                         |

**Figure 1.** (a) Temporal changes of the annual global mean surface air temperature (SAT; unit: °C) (blue) and CO<sub>2</sub> concentration (unit: ppm) (red). The solid lines represent the ensemble means, while the shaded areas indicate the 25th to 75th percentile range across 28 members. Three dashed gray lines mark the years 2140, 2220, and 2280, highlighting important temporal milestones in the experiment. (b) Temporal changes of annual-mean terrestrial near-surface wind speed (NSWS; unit: m s<sup>-1</sup>) over Northern Hemisphere (20°N–70°N) (orange) and North Atlantic (0°N–70°N, 60°W–30°E) (dark blue). Anomalies are calculated relative to the average NSWS of the constant CO<sub>2</sub> scenario. An 11-year running mean has been applied to smooth out inter-annual variability.

**Figure 2.** (a) Decadal trend in annual-mean near-surface wind speed (unit: m s<sup>-1</sup> decade<sup>-1</sup>) during the CO<sub>2</sub> ramp-up period (2001–2140). Grid points with shadings denote the tendencies are significant at the 0.05 level. (b–d) Same as (a), but for tendencies during the CO<sub>2</sub> ramp-down period (2141–2280), early CO<sub>2</sub> ramp-down period (2141–2220), and late CO<sub>2</sub> ramp-down period (2221–2280), respectively.

**Figure 3.** (a) Temporal changes of the Atlantic Meridional Overturning Circulation index (unit: Sv). The solid line represents the ensemble mean, and the shaded area shows the 25th to 75th percentile range of 28 ensemble members. Dashed gray lines at 2140, 2220, and 2280 denote significant temporal markers. (b–e) Same as (a), but for Northern Hemisphere (NH) surface air temperature gradient (0°N–30°N minus 60°N–90°N; in °C), NH westerly jet strength (the average for 30°N–60°N; in m s<sup>-1</sup>) at 500 hPa, North Atlantic (NA, 60°W–30°E) surface air temperature gradient (0°N–30°N minus 60°N–90°N; in °C), and NA westerly jet strength (the average for 30°N–60°N, 60°W–30°E; in m s<sup>-1</sup>) at 500 hPa, respectively. An 11-year running mean has been applied to smooth out inter-annual variability.

**Figure 4.** (a) Inter-ensemble correlation coefficients between the 28 ensemble members: the AMOC versus North Atlantic (NA, 60°W–30°E) surface air temperature (SAT) gradient (60°N–90°N minus 0°N–30°N) from 2000 to 2300. Three dashed gray lines denote 2140, 2220 and 2280, respectively. Two dashed blue lines denote the 0.05 significance level. (b) Same as (a), but for correlation coefficients between the AMOC and NA westerly jet strength (30°N–60°N, 60°W–30°E) at 500 hPa. (c) Same as (a), but for correlation coefficients between the AMOC and terrestrial near-surface wind speed (NSWS) over Europe (30°N–60°N, 5°W–60°E). An 11-year running mean has been applied to smooth out inter-annual variability.

**Figure 5.** The physical mechanisms by which CO<sub>2</sub> and Atlantic Meridional Overturning Circulation modulate extratropical near-surface wind speed during three periods.

**Figure 6.** (a) Changes of terrestrial annual-mean near-surface wind speed over Europe (30°N–70°N, 5°W–50°E) as a function of CO<sub>2</sub> concentrations after 11-year running mean. The rampup (dark blue), ramp-down (orange), and stabilization (yellow) are denoted with different colors. The solid line represents the ensemble mean, and the shaded area shows the 25th to 75th percentile range of 28 ensemble members. (b-c) Same as (a), but for Asia (20°N–70°N, 60°E–180°E) and North America (30°N–70°N, 170°W–50°W). The range of Y-axis is 0.25 m s<sup>-1</sup> in each subplot.