# Peer review of "Asymmetric response of Northern Hemisphere near-surface wind speed to"

_EGUsphere, 2025_

## Author Comment (AC1)

**Response to the Reviewers' Comments**

Dear editor and reviewers,

We greatly appreciate the reviewers' constructive comments. Our responses to the reviewers' feedback are outlined below. All reviewers' comments are presented in black, while the authors' responses are in blue. Please note that all line numbers referenced in this response letter correspond to those in the **clean** version of the revised manuscript.

**Responses to Reviewer #1**

**General Comments:** Based primarily on CESM1.2 large ensemble results, this manuscript investigates the evolution of near-surface wind speed (NSWS) in the Northern Hemisphere under varying $CO_2$ concentrations. It reveals the particular behavior of NSWS over Europe and explores the possible underlying causes. What interests me most is the finding that the Atlantic Meridional Overturning Circulation (AMOC) plays a key role in driving the evolution of NSWS over Europe in response to $CO_2$ changes, mainly by modifying the temperature gradient over the Atlantic. The analysis is sound, and the logical flow is clear. To effectively disseminate these new findings, I believe this manuscript is worthy of publication in this journal.

We are grateful for the reviewer's high compliments on our paper. We sincerely thank the reviewer for the constructive comments, which greatly helped us to improve this study. We have revised carefully the manuscript following the comments and suggestions. And we believe that all issues have been adequately addressed in the revised version of the manuscript.

**Specific Comment 1**: A large portion of the manuscript discusses NSWS changes across the entire Northern Hemisphere, while the title focuses specifically on Europe. It would be better to unify the scope; either adjust the title or concentrate the discussion more on Europe.

As suggested, we revised the title to "Asymmetric response of **Northern Hemisphere** near-surface wind speed to $CO_2$ removal".

**Specific Comment 2**: Both the CESM large ensemble and CDRMIP experiments have limitations. The former may suffer from model dependency, while the latter may be affected by internal variability due to the limited number of models involved. Therefore, it remains uncertain whether NSWS would behave exactly under CO2 removal scenario as presented in this manuscript. However, the results do robustly suggest that AMOC is the primary driver of the NSWS evolution.

We fully acknowledge the limitations pointed out by you. These uncertainties imply that the exact NSWS response under the $CO_2$ removal scenario presented in our study could vary somewhat with different modeling frameworks or additional ensemble members. We have clarified these limitations explicitly in the revised manuscript and suggested that future

studies employ more comprehensive multi-model ensembles by adding the following text to Section 5 in the revised manuscript:

 "*We acknowledge several uncertainties inherent in our findings, including potential model-dependency arising from CESM's large ensemble experiment and limited realizations from the CDRMIP simulations. Future studies involving more extensive multi-model ensembles could further validate and enhance the robustness of our conclusions*".

**Responses to Reviewer #2**

**General Comments:** This study investigates how near-surface wind speed (NSWS) across NH land areas responds to CO2 ramp-up and ramp-down. Increasing CO2 concentrations are shown to lead to an overall decrease in NSWS. When CO2 levels decrease, NSWS quickly recovers within the first several decades but then stall until the end of CO2 removal. NSWS then declines again, particularly over Europe, in the initial stabilization period, followed by a slow recovery in the mid to late stabilization period. These non-monotonic changes in NSWS are attributed to the AMOC changes and the associated changes in temperature gradients and westerly jets. This finding underscores the pivotal role of the AMOC in shaping the hysteresis of NSWS in CDR experiment.

Though short and simple, this study addresses an interesting subject not covered in previous studies. The results of this study are not surprising and can be inferred from previous studies (e.g., An et al. 2021). Nevertheless, it is beneficial to quantify it. In this regard, I appreciate this study. However, I found substantial room for improvement in the study. Among others, the possible mechanism(s) is too simple and not justified in quantity. I suggest that the authors address the following issues when revising the manuscript.

We thank the reviewer for providing useful suggestions and insightful comments for us to improve this study. Specifically, we have provided a more comprehensive and quantitative justification of the underlying mechanisms. Detailed responses to each comment are presented below.

**Major Comment 1 (Mechanism)**: NWSW is explained for both NH and NA. However, only the changes over NH are presented in Fig. 1b. It would be helpful to show the temporal evolution of NA NSWS in a new panel below Fig. 1b. This would be particularly useful when discussing Fig. 3.

Your points are well received. Following your suggestions, we have included these updated figures (attached below as Fig. R1) and detailed discussions in the revised manuscript (*Lines 230–238*).

[Figure]

**Figure R1.** (a) Temporal changes of the annual global mean surface air temperature (SAT; unit: °C) (blue) and $CO_2$ concentration (unit: ppm) (red). The solid lines represent the ensemble means, while the shaded areas indicate the 25th to 75th percentile range across 28 members. Three dashed gray lines mark the years 2140, 2220, and 2280, highlighting important temporal milestones in the experiment. (b) Temporal changes of annual-mean near-surface wind speed (unit: m s$^{-1}$) over Northern Hemisphere land (20°N–70°N) (orange) and North Atlantic (0°N–70°N, 60°W–30°E) (dark blue). Anomalies are calculated relative to the average NSWS of the constant $CO_2$ scenario. An 11-year running mean has been applied to smooth out inter-annual variability.

NSWS changes are mainly explained by SAT gradient changes. However, SAT gradient does not explain everything. An example is Fig. 3. While AMOC continues to weaken from the ramp-up to early ramp-down periods (Fig. 3a), NA SAT gradient increases during the ramp-up period and then rapidly decreases (Fig. 3d). This SAT gradient changes do not correspond to NA westerly jet  changes (and presumably NA NSWS changes as well), especially during the ramp-up period (Fig. 3e). Can you explain why?

Another example is Figs. 1b and 3. Fig. 1b shows a rapid weakening of NH NSWS during the initial stabilization period, followed by a steady decrease afterward. This evolution cannot be explained by NH SAT gradient change shown in Fig. 3b. This mismatch indicates that other factors also affect NSWS changes. Such factors should be discussed in detail.

We apologize for the confusion regarding the mismatch, which arose mainly due to the definition and sign convention of the meridional SAT gradient in the original manuscript. Previously, we defined the gradient as the SAT **meridional** difference (high minus low latitude), producing negative values that could misleading suggest upward trends when gradients weakened. To clarify, we have now clearly redefined the SAT gradient as tropical SAT minus high-latitude SAT, ensuring that weakening gradients correspond explicitly to downward trends in our **revised** figures. Accordingly, the updated Fig. 3 (shown as Fig. R2 below) can demonstrate decreasing NA (Fig. R2d) and NH (Fig. R2b) SAT gradients during the ramp-up period. We believe the modification resolves the confusion and ensures consistently with the evolution of the AMOC and westerly jet.

Specifically, to clarify the underlying physical mechanisms, we provide the following detained explanations in the revised manuscript:

(1) During the ramp-up period, two competing influences operate over the NA region. The weakening AMOC reduces northward heat transport, which by itself would enhance the NA SAT gradient. In contrast, the globally rising $CO_2$ levels strongly warm high latitudes relative to the tropics, significantly weakening the large-scale SAT gradient. In net, the warming-driven gradient weakening effect dominates, clearly resulting in decreased NA SAT gradient (Fig. R2d), decreased NA westerly jet intensity (Fig. R2e), and ultimately decreased NA NSWS (Fig. R1b). Thus, despite the weakening AMOC trend, its modest magnitude cannot counterbalance the strong, $CO_2$-driven SAT gradient decrease.

During the early ramp-down period, however, the situation changes. Continued weakening of the AMOC now acts in concert with global cooling due to $CO_2$ removal. Both mechanisms reinforce a strong SAT gradient increase, leading to coherent, rapid enhancements of the NA SAT gradient (Fig. R2d), the NA westerly jet (Fig. R2e), and NA NSWS (Fig. R1b).

(2) During the initial stabilization period, the GMST shows a stabilization while the AMOC shows an overshoot enhancement. The enhanced AMOC would transport anomalous warmer water to the north, which favors weakened SAT gradient, and weakened NH and NA NSWS (Fig. R1b). During the later stabilization period, both the GMST and AMOC show slightly decreasing trends. Together, they lead to slightly increasing trends of NH and NA SAT, so the NH and NA NSWS show slightly increasing trends (Fig. R1b).

[Figure]

**Figure R2.** (a) Temporal changes of the Atlantic Meridional Overturning Circulation index (unit: Sv). The solid line represents the ensemble mean, and the shaded area shows the 25th to 75th percentile range of 28 ensemble members. Dashed gray lines at 2140, 2220, and 2280 denote significant temporal markers. (b–e) Same as (a), but for Northern Hemisphere (NH) surface air temperature gradient (0°N–30°N minus 60°N–90°N; in °C), NH westerly jet strength (the average for 30°N–60°N; in m s⁻¹) at 500 hPa, North Atlantic (NA, 60°W–30°E) surface air temperature gradient (0°N–30°N minus 60°N–90°N; in °C), and NA westerly jet strength (the average for 30°N–60°N, 60°W–30°E; in m s⁻¹) at 500 hPa, respectively. An 11-year running mean has been applied to smooth out inter-annual variability.

**Major Comment 2 (Model biases)**: It is stated that NSWS climatology in each model is similar to ERA5 climatology. The spatial correlation over 60°S–70°N (not just NH land areas) ranges from 0.72 to 0.85 (L125). Is this also the case when considering only NH land areas? Beyond the spatial distribution, does the model reproduce the intensity of NSWS? I suggest presenting NWSW climatology for ERA5 and each model in the SI.

Good suggestion, we have added the climatology of ERA5 and each model as Fig. S1 in the supplementary information (attached below as Fig. R3 for your reference). Additionally, we quantified the pattern correlation coefficients between each model and ERA5 climatologies focusing exclusively on NH land areas (20°N–70°N). The resulting correlation coefficients are 0.71 (CESM1.2), 0.59 (CanESM5), 0.52 (MIROC), and 0.83 (NorESM2), indicating that the reasonable spatial agreement of NSWS distributions between the models and ERA5.

Moreover, we assessed the intensity differences between model-simulated NSWS and ERA5 observations. Specifically, the weighted area root mean square difference over the NH terrestrial region are 1.03, 1.07, 1.37, and 0.71 m s⁻¹ for CESM1.2, CanESM5, MIROC, and NorESM2, respectively. Thus, the NSWS magnitudes simulated by these models are also comparable with ERA5. This quantitative analysis, along with a detailed description, has been included in the revised manuscript (***Lines 126–138***).

[Figure]

**Figure R3.** (a) Climatology (1979–2018) annual-mean near-surface wind speed (unit: m s⁻¹) from the ERA5. (b–e) Climatology (the first 100 years of the piControl experiment) annual-mean near-surface wind speed from the CESM 1.2, CanESM5, MIROC, and NorESM2, respectively.

**Major Comment 3 (Comparison to CDRMIP):** Figs. 1b and S2 reveal a notable difference in NSWS changes among the models. These differences are qualitatively related to the different AMOC changes. Could you quantify it by establishing NSWS-AMOC relationship among the models? Since NSWS changes resemble 500-hPa wind changes, 500-hPa winds could be also used to increase the sample size.

We agree with your observation. To quantitatively evaluate the role of the AMOC in these differing NSWS responses, we performed a **bi-linear regression** analysis of NH extratropical NSWS on GMST and AMOC for each of the three CDRMIP models (CanESM5, MIROC,

and NorESM2). Specifically, we divided the 280-year simulations into two equal-length periods: the ramp-up period (years 1–140) and the ramp-down period (years 141–280).

Our analysis (summarized in Table R1 below) show that GMST consistently explains over 90% of NSWS changes across all models and both periods, confirming a generally weak AMOC influence. These findings align well with CESM, where GMST dominates NSWS changes during the ramp-up and early ramp-down phases. And only when the AMOC anomalies become substantial, as seen in CESM's late ramp-down period, does AMOC's influence become dominant. Therefore, it is inappropriate to fit a simple linear relationship between NSWS and AMOC when the AMOC contribution remains minimal, and the same reasoning also applies to the 500 hPa wind. To clarify, we have added "*Besides, we also performed similar analysis for the three CDRMIP models. Among three models and two periods, the contributions of GMST always dominate (contributions >90%) the NH extratropical NSWS changes, further supporting that a substantial AMOC anomaly is required for AMOC to notably influence NSWS changes, as observed in CESM*" in the last paragraph of Section 4 (***Lines 280–284***).

**Table R1**. Relative contributions of GMST and AMOC to NH extratropical NSWS changes from three CDRMIP models during the ramp-up and ramp-down periods.

|  | Ramp-up period | | Ramp-down period | |
|---|---|---|---|---|
|  | GMST | AMOC | GMST | AMOC |
| CanESM5 | 92.3% | 6.8% | 95.1% | 3.6% |
| MIROC | 96.6% | 3.1% | 94.4% | 5.2% |
| NorESM2 | 95.2% | 4.1% | 92.7% | 6.3% |

**Major Comment 4 (Cross-member correlation)**: I am not sure what the purpose of this analysis is. Ensemble spread of AMOC is rather small during the ramp-up period and is unlikely to affect ensemble spreads of NA SAT gradients and westerly jets. As ensemble spread increases during the ramp-down period, AMOC more effectively explains ensemble spread of NA climate properties. This does not suggest "the crucial effect of AMOC recovery on weakening the NA SAT gradient" in L248-L249. Rather it suggests that the internal variability (not trend!) of NA climate properties is closely associated with AMOC variability during the ramp-down period.

We appreciate the reviewer's insightful comment regarding the interpretation of the cross-member correlation analysis. Indeed, we fully agree that the cross-member correlations presented in Figure 4 primarily reflect internal variability rather than externally forced trends. Our original phrasing ("crucial effect") might have inadvertently suggested a forced influence rather than emphasizing internal variability, potentially causing confusion.

To explicitly clarify our intention and avoid such misunderstanding, we have substantially revised the relevant paragraph. The purpose of this analysis is to highlight that internally generated AMOC variations can significantly influence the internal variability of NA climate properties, especially when the ensemble spread is large, as observed during the late ramp-down period. Specifically, we have revised the paragraph as follows:

"*Internally generated AMOC changes can also support our argument about the potential roles played by CO2-forced AMOC changes. To quantify how internal AMOC variability contributes to the cross-member spread of key NA climate variables, Figure 4 shows the yearly Pearson correlation coefficients among the 28 CESM ensemble members between AMOC strength and (i) the NA meridional SAT gradient, (ii) the NA westerly-jet intensity, and (iii) the European (30°–60 °N, 5 °W–60 °E) NSWS. Because all members share identical external forcing, these inter-ensemble correlations isolate internal variability. Significant correlations thus indicate that internal AMOC fluctuations are major drivers of atmospheric variability among ensemble members. During the ramp-up period (2001–2140), the ensemble spread of AMOC remains small, and correlations with the three metrics are weak and statistically insignificant. However, during the late ramp-down period (2221–2280), the spread of AMOC substantially increases, and the correlations become statistically significant. These year-by-year correlations clearly demonstrate that larger (smaller) AMOC anomalies are associated with a weaker (stronger) NA SAT gradient, a reduced (enhanced) westerly jet, and lower (higher) European NSWS.*" We have incorporated these revisions into the manuscript (***Lines 256–270***).

**Major Comment 5 (Bi-regression analysis)**: I had a difficult time understanding L253-L257: "During the ramp-up period (2001–2140), GMST and AMOC explain the variance of NH extratropical NSWS for 98% and 1.2%, respectively. During the early ramp-down period (2141–2220), GMST and AMOC explain 95.3% and 4.2% of the variance, respectively. While during the late ramp-down period (2221–2280), GMST and AMOC explain 24.5% and 73.2% of the variance, respectively". Does this mean that AMOC is not important for NSWS changes during the ramp-up and early ramp-down periods? If so, this contradicts the key conclusion of the paper – the critical role of AMOC hysteresis in NSWS hysteresis.

We agree that AMOC contributes only marginally to NH extratropical NSWS variance during the ramp-up (1.2%) and early ramp-down (4.2%) phases. To clarify explicitly, our central conclusion is that the influences of GMST and AMOC on NSWS are phase dependent. During the late ramp-down period, the AMOC rapidly rebounds, and the effect of GMST is lower than that during the early ramp-down period. Thus, the bi-linear regression shows AMOC's dominance (73.2%) over GMST (24.5%). This quantitative finding aligns consistently with our overarching message in Abstract, emphasizing that a sufficiently strong AMOC recovery can override the GMST-controlled response of the westerly jet and NH extratropical NSWS. To eliminate confusion and explicitly highlight this phase dependence, we have substantially revised the first paragraph of Section 5 as follows:

*In this study, we utilized a CESM1.2 CO₂ removal experiment to evaluate the response of NH extratropical NSWS to anthropogenic CO₂ emission and subsequent removal. Our analyses reveal an asymmetric response of NSWS during the ramp-up (2001–2140) and ramp-down (2141–2280) periods, driven by phase-dependent interactions between GMST changes and AMOC-related heat transport anomalies. Figure 5 summarizes the underlying physical mechanisms: During the ramp-up period (Figure 5a), the gradually weakening AMOC transports less warm water northward, thereby enhancing the meridional SAT gradient and*

*thus strengthen NH and NA NSWS. However, the counteracting effect of $CO_2$-driven global warming strongly reduces the NH meridional SAT gradient, ultimately dominating the response and resulting in decreased NSWS across NH extratropics. During the early ramp-down period (Figure 5b), $CO_2$ removal-induced global cooling combines with further AMOC weakening to substantially enhance the SAT gradient, intensifying the westerly jet and increasing NH NSWS, with GMST remaining the dominant driver due to its stronger anomalies. Conversely, during the late ramp-down period (Figure 5c), a rapid AMOC recovery substantially increases northward heat transport, significantly weakening the NA SAT gradient and subsequently diminishing the westerly jet and NSWS. This AMOC-driven weakening outweighs concurrent weaker GMST cooling-induced strengthening of the SAT gradient, inducing a weakening NH and NA NSWS (**Lines xxx–xxx**).*

**Minor Comment 1**: Fig. 4 is not critical and could be moved to the SI.

Point taken. We have moved this figure to SI as suggested.

**Minor Comment 2**: Fig. 5 should be extended to the year 2500 (i.e., the end of stabilization). The same applies to Fig. 7.

Good suggestion. We have extended the X-axis to 2500 in these figures (Figs. R4 and R5 below).

[Figure]

**Figure R4.** (a) Changes of terrestrial annual-mean near-surface wind speed over Europe (30°N–70°N, 5°W–50°E) as a function of $CO_2$ concentrations after 11-year running mean. The ramp-up (dark blue) and ramp-down (orange) are denoted with different colors. The solid line represents the ensemble mean, and the shaded area shows the 25th to 75th percentile range of 28 ensemble members. (b-c) Same as (a), but for Asia (20°N–70°N, 60°E–180°E) and North America (30°N–70°N, 170°W–50°W). The range of Y-axis is 0.25 m s$^{-1}$ in each subplot.

[Figure]

**Figure R5.** (a) Changes of terrestrial annual-mean near-surface wind speed over Europe (30°N–70°N, 5°W–50°E) as a function of $CO_2$ concentrations after 11-year running mean. The ramp-up (dark blue), ramp-down (orange), and stabilization (yellow) periods are denoted with different colors. The solid line represents the ensemble mean, and the shaded area shows the 25th to 75th percentile range of 28 ensemble members. (b-c) Same as (a), but for Asia (20°N–70°N, 60°E–180°E) and North America (30°N–70°N, 170°W–50°W). The range of Y-axis is 0.25 m s$^{-1}$ in each subplot.

---

## Author Response (AR2)

**Response to the Editor's Comments**

General Comments: Thank you for your detailed revision and reply to the reviewers' comments, through which the quality of the manuscript has been substantially improved. In the second round of review, one referee suggests acceptance of the manuscript. The other referee who raised some major issues of the manuscript declined to review the revision, thus I did the review by myself. Based on the second round of review and my own evaluation, the manuscript is possible to be acceptable for publication after moderate revisions. The authors need to do more work to clarify or discuss the mechanism for the change of the near-surface wind speed, given the article scope of the WCD. Here are my suggestions on paper revision for consideration.

We are grateful for the Editor's constructive comments. We have revised carefully the manuscript following the comments and suggestions.

**Comment 1**: Mechanism for the non-monotonic change of the near-surface wind speed (NWSP)

In authors' response to Referee 2 and in the revised manuscript, the authors argue that the different relationship between the AMOC and the NWSP in Ramp-up and Ramp-down periods may attribute to the two somewhat competing influences: changes of AMOC and the change of GMST (global mean surface temperature). However, the variation of AMOC can also influence the planetary scale surface temperature, which can be an important part of the GMST. Though the authors have conducted bi-regression analysis to show their relative importance, these two effects are not independent. The authors at least need to point out this during the discussion.

As suggested, we added the following text to the second paragraph of Section 5 in the revised manuscript: "Besides, the effects of AMOC and GMST are not completely independent due to the variation of AMOC also influencing large-scale surface temperature. Essentially, it is the Arctic Amplification and AMOC's competing." (Lines 329–331)

Comment 2: The relationship between the North Atlantic jet speed and the surface wind speed

From Figure 3, in the ramp-up period, the relationship between the North Atlantic jet speed and the NWSP is different from other periods. Please note that in this period, the Northern Hemispheric (NH) and North Atlantic (NA) surface wind speed, SAT gradient and the AMOC all show decreasing trends, except for the NA jet speed. I suggest the authors check this result (e.g. check the change of jet speed in other vertical levels or whether the result depends on the definition of North Atlantic). If the surface wind and surface SAT gradient over North Atlantic all decrease, it is not clear to me why the jet speed increase. Is there any explanation for this?

Your observation is correct. Our previous analysis neglected to describe and discuss this detail. Because we wanted to consider the combined large-scale climate effects of the AMOC and CO2, the current definition of the NA meridional temperature gradient uses 0–30° as low latitudes and 60–90° as high latitudes. However, the spatial distribution of surface air

temperature trends during the CO2 ramp-up period (Fig. S2a) reveals the presence of a cold blob in the NA, with a region of warming to its south. For the mid-latitudes of the NA, this regional surface temperature change would strengthen the north-south meridional temperature gradient. According to thermal wind relations, the NA westerly jet would strengthen (Fig. 3e). On a larger scale, the Arctic amplification caused by global warming due to CO2 increases weakens the north-south meridional temperature gradient (both globally and in the NA). Therefore, the changes in the meridional temperature gradient caused by AMOC and CO2 offset each other, and in the confrontation, AMOC has a stronger climate effect on the intensity of the NA westerly jet. Hence, the NA westerly jet is slightly strengthened, and NSWS over Europe is slightly strengthened (Fig. 2a). However, in the entire NH, the climate effect of CO2 is stronger, so the NH meridional temperature gradient and the NH westerly jet are weakened synchronously (Figs. 3b–c).

We added the following discussions in Section 4 as follows:

"However, a slight increase in the NA westerly jet is observed in Fig. 3e. This is due to an AMOC-associated cold blob and a warming center to its south (Fig. S2a), which enhances the regional meridional temperature gradient, and its effect on the NA westerly jet is stronger than that from the CO2 increase. Hence, the NA westerly jet is slightly strengthened, and NSWS over Europe is slightly strengthened (Fig. 2a)." (Lines 242–246)

In addition, the authors need to clarify the definition of NH and NA near-surface wind speed. According to the authors' statement in section 2.4 in Line 145, all the calculations of near-surface wind speed focus on land regions. Does this rule also apply to the authors' calculation of the NA near-surface wind speed? And do the authors' calculations of the global and regional mean surface temperature apply for both land the oceanic surface? Those key information need to be clarified.

Thanks for this suggestion. NA NSWS is calculated only over the ocean. Surface temperature is calculated over both ocean and land. We clarified these details in Section 2.4:

"All calculations related to NSWS are focused on the land regions, except that North Atlantic NSWS focuses on oceanic NSWS. Calculations related to surface temperature include both ocean and land regions." (Lines 147–149)

**Comment 3: Cause-effect issue for the role of AMOC**

I agree that the AMOC can be an important player for the change of the near-surface wind. However, the authors' current argument seems to claim that the AMOC is a main reason for the decrease of the surface wind, especially in the Ramp-down period. But what's the reason for the change of AMOC? Does the change of the surface wind also affect the AMOC, especially in the Ramp-down and Stabilization periods in which the initial state of the AMOC may have little influence on the AMOC evolution (Is there any initialization of the ocean state before the Ramp-up run?). Is there any significant lead-lag correlation between AMOC, SAT gradient and surface wind?

We appreciate the reviewer's insightful question.

- (i) The reason for the change of AMOC: The AMOC variation is a buoyancy-driven response with strong salt-advection feedback and stratification changes. During the ramp-up period, CO2-forced surface heat/freshwater fluxes freshen and warm the subpolar North Atlantic, weakening deep convection and AMOC. During the early ramp-down period, due to the slow recovery of AMOC, it shows strong hysteresis. During the late ramp-down period, the subtropic–subpolar salinity gradient and reduced stratification favor enhanced densewater formation and an AMOC "overshoot"/rapid recovery. This mechanism and timing have been diagnosed in detail for the same experimental design by An et al. (2021), who showed that subpolar North Atlantic temperature lags the AMOC by 10–20 years and that the AMOC overshoot arises from amplified salt-advection feedback under evolving stratification and surface freshwater fluxes.
- (ii) Does the change of NSWS affect AMOC: In principle, wind stress and its curl can modulate gyre circulation and thus the overturning. However, in this particular CO₂-pathway experiment, the multi-decadal AMOC evolution is dominated by buoyancy forcing and salt-advection feedback rather than by wind-driven changes. An et al. (2021) explicitly traced the AMOC tendency to salinity advection and surface freshwater/heat flux anomalies. They showed AMOC leads the subpolar temperature by 10–20 years (AMOC → subpolar SST/stratification → atmosphere). In our analysis, when AMOC anomalies become large during the late ramp-down period, inter-ensemble correlations show that members with stronger AMOC are systematically associated with a weaker NA meridional SAT gradient, a reduced NA westerly jet, and lower European NSWS (Fig. 4), again consistent with AMOC forcing the atmosphere. In these simulations, Ramp-down and Stabilization are a seamless continuation of Ramp-up, so the AMOC trajectory in these periods reflects the thermohaline anomalies accumulated along the pathway rather than sensitivity to the present-day initial phase. Under these conditions, any feedback from NSWS onto the AMOC is secondary relative to buoyancy control.
- (iii) Is there any initialization of the ocean state before the Ramp-up run: Yes. CESM1.2 was first integrated for 900 years at fixed 367 ppm, which is a present-day (PD) state, to generate an equilibrium ocean—atmosphere state. The 28 ramp-up members then branched from distinct PD-year ocean/atmosphere initial conditions that sample different phases of multidecadal variability (AMO/PDO). The ramp-down run continues directly from the end of ramp-up following the CDRMIP symmetry protocol. We revised the related description into "and the model is integrated for 900 years to generate an equilibrium ocean—atmosphere state" in Lines 116–117.
- (iv) Is there any significant lead-lag correlation between AMOC, SAT gradient and surface wind: We agree that significant lead-lag relationships are expected: in the North Atlantic, AMOC tends to lead subpolar SST and the regional meridional SAT gradient by roughly 10–20 years, as well as atmospheric wind/jet responses. In our study, the three analysis periods span 140, 80, and 60 years, respectively; within such multi-decadal windows, a 10–20-year lead is effectively absorbed by the period-mean and year-by-year zero-lag cross-member diagnostics we use to isolate the forced, contemporaneous covariability. Moreover, under a non-stationary CO2 pathway, the lead is state-dependent rather

than constant, so fixed-lag correlations can be less informative than contemporaneous relationships for attribution.

In the revised manuscript, we acknowledge the expected decadal lead–lag chain (AMOC → SAT gradient & westerly jet & NSWS), while noting that our conclusions are based on zero-lag cross-member covariation that remains robust to such decadal offsets. A full state-dependent lead–lag quantification is beyond our present scope. Specifically, we added "Although AMOC is expected to lead subpolar NA SST and the regional meridional SAT gradient by 10–20 years (An et al., 2021), our analysis is based on zero-lag cross-member covariation over multi-decadal windows. Such decadal leads are small relative to these windows and do not alter the sign or interpretation of the contemporaneous relationships reported here. We therefore emphasize contemporaneous, forced covariation for attribution, while acknowledging the expected lead–lag behavior." in Lines 292–297.

**References:**

An, S.-I., J. Shin, S.-W. Yeh, S.-W. Son, J.-S. Kug, S.-K. Min, and H.-J. Kim (2021). Global cooling hiatus driven by an AMOC overshoot in a carbon dioxide removal scenario, Earth's Future, 9, e2021EF002165. <a href="https://doi.org/10.1029/2021EF002165">https://doi.org/10.1029/2021EF002165</a>

**Comment 4**: Figure 5**

The schematic of Figure 5 does not accurately represent the results (i.e. Figures 3 and 4). For example, in Figure 4, in the Ramp-up and Stabilization periods, there is no significant correlation between AMOC index and the NA westerly jet. Though the figure mainly reflects the internal variability, the relationship between AMOC and the atmosphere is supposed to exist for decadal and longer time scales. Therefore, the change of the westerly jet in Figure 5a with the AMOC is not convincing to me. The correlation between the AMOC and the SAT gradient seems more robust and direct. Please note that the 500 hPa jet stream and the surface wind do not have to change in the same way. In the North Atlantic, the surface wind strength is usually more related to the eddy-driven jet, which is further related to the eddy activity and background baroclinicity.

We revised the anomalous warming center in Fig. 5a. Also, we replaced the westerly jet with a westerly jet trend in the schematic figure, with a solid line representing an enhanced jet and a dashed line representing a weakened jet, respectively.

**Fig. 5.** The physical mechanisms by which CO2 and Atlantic Meridional Overturning Circulation modulate extratropical near-surface wind speed during three periods. The warmer and colder centers denote the combination effects of GMST and AMOC. The strengthened NA westerly jet is in solid line, while the weakened jet is in dashed line.

**Comment 5**: Abstract**

The authors' argument on the role of AMOC in line 43-44 ("phase dependent role") as well as in the concluding part is not very clear and maybe misleading, as it is very easily to link to the phase of AMOC. The results actually suggest that the AMOC's effect is different in different periods of CO2 variation, with most dominant influence in the late Ramp-down period.

Agree. We deleted the descriptions about "phase dependent" in the Abstract and Section 5.

---

## Author Response (AR3)

**Response to the Editor's Comments**

Comments: Thank you for your detailed response to my comments in the last review and revising the manuscript accordingly, which clarified several key issues in the study. I think the current format of the manuscript is almost ready for publication after some small edition. I strongly suggest the authors to move Figures S2 and S6 into the main body of the manuscript, given that those two figures have been greatly discussed in the text and are important for understanding some main conclusions of the paper. Currently, there are six figures in the main text and there is still room for more figures. Congratulations in advance.

We are grateful for the Editor's comments. We have moved the Figs. S2 and S6 into the main body of the manuscript.